# Assessment of Groundwater Potential Zones Using GIS and Fuzzy AHP Techniques—A Case Study of the Titel Municipality (Northern Serbia)

Mirjana Radulović [1,*], Sanja Brdar [1], Minučer Mesaroš [2], Tin Lukić [2], Stevan Savić [2], Biljana Basarin [2], Vladimir Crnojević [1] and Dragoslav Pavić [2]

1   BioSense Institute—The Research and Development Institute for Information Technologies in Biosystems, Zorana Đinđića 1, 21000 Novi Sad, Serbia; sanja.brdar@biosense.rs (S.B.); crnojevic@biosense.rs (V.C.)

2   Department of Geography, Tourism and Hotel Management, Faculty of Sciences, University of Novi Sad, Trg Dositeja Obradovića 3, 21000 Novi Sad, Serbia; minucer.mesaros@dgt.uns.ac.rs (M.M.); tin.lukic@dgt.uns.ac.rs (T.L.); stevan.savic@dgt.uns.ac.rs (S.S.); biljana.basarin@dgt.uns.ac.rs (B.B.); dragoslav.pavic@dgt.uns.ac.rs (D.P.)

\*   Correspondence: mirjana.radulovic@biosense.rs; Tel.: +381-(69)511-0235

**Abstract:** Groundwater is one of the most important natural resources for reliable and sustainable water supplies in the world. To understand the use of water resources, the fundamental characteristics of groundwater need to be analyzed, but in many cases, in situ data measurements are not available or are incomplete. In this study, we used GIS and fuzzy analytic hierarchy process (FAHP) techniques for delineation of the groundwater potential zones (GWPZ) in the Titel Municipality (northern Serbia) based on quantitative assessment scores by experts (hydrologists, hydrogeologists, environmental and geoscientists, and agriculture experts). Six thematic layers, such as geology, geomorphology, slope, soil, land use/land cover, and drainage density were prepared and integrated into GIS software for generating the final map. The area falls into five classes: very good (25.68%), good (12.10%), moderate (15.18%), poor (41.34%), and very poor (5.70%). The GWPZ map will serve to improve the management of these natural resources to ensure future water protection and development of the agricultural sector, and the implemented method can be used in other similar natural conditions.

**Keywords:** groundwater; geographic information systems (GIS); water management; fuzzy analytic hierarchy process (FAHP)

## 1. Introduction

Groundwater represents one of the most important natural resources collected in the subsurface geological structures [1]. It is the largest available freshwater resource on the earth, which primarily serves domestic purposes, industrial, and agricultural uses [2–4]. According to FAO [5], groundwater represents 26% of global renewable freshwater resources. Due to the drastic population growth but also the global impact of climate change, repetitive drought conditions, and lack of rainfall [6], demand for groundwater resources has been drastically increasing in the past decades. In recent years, increasing agriculture production is putting pressure on groundwater resources for irrigation, especially in arid and semi-arid regions, which require an enormous amount of water supply and are usually used unsustainably. Further, it causes problems, such as decreased groundwater levels, water pollution from agriculture, and degradation of water quality [7,8]. To manage these problems, we have to know how much water exists underground and how much we can use for different purposes. Groundwater monitoring is very important when it comes to understanding the capacity and possibilities of existing reservoirs. Data can help us understand natural and artificial factors, which could affect groundwater level, quality, as well as consumption [9,10]. Based on new knowledge and research, governments could

pay more attention to developing a sustainable groundwater management practice, and planning delineation of the groundwater potential zones is among the first steps for creating management strategies.

Groundwater in Vojvodina is one of the most important natural resources, especially in agriculture, presenting the main source for crop growth. Collected in the Neogene and Quaternary sediments, groundwater in Serbia is abundantly distributed, where 75% of the population is using these resources for a different purpose. Knowledge about groundwater regime and capacity is important for irrigation and land drainage. Despite this fact, research about the availability and useability of groundwater in Serbia is deficient and has not been sufficiently studied. According to the research of Jaroslav Černi Water Institute, only 30% of groundwater is used out of the total estimated of 67 $m^3$/s [11]. However, unsustainable water exploitation from the principal aquifer consuming more than the rate of restoration caused a significant decrease in the piezometric level in some regions in Serbia [11]. Based on that, it is very important to work on the identification of the potential location for new groundwater development. The aim is to improve the sustainable management of groundwater resources in Vojvodina region, the main agricultural region in Serbia, to conduct groundwater exploration faster and more efficiently.

With the advent of digital technology, including powerful computers and the possibility for integration of modern methods within a geographical information system (GIS) and remote sensing, it is possible to work on more complex and advanced hydrological research, such as delineation of groundwater potential zones (GWPZ), in any region [12]. These advantages replaced traditional data collecting and enabled easier and cheaper access to data [13]. Using GIS techniques, it is possible to operate huge databases, which are crucial for creating a good system for the decision-making process in different fields [3,14]. Researchers are now familiar with geospatial technology, which is necessary for generating spatial results, enabling more successful and easier processes for decision makers. Using factors that control the occurrence, origin, and movement of groundwater, such as geology, geomorphology, slope, soil, land use/land cover, drainage density, rainfall intensity, anthropogenic factors, etc., researchers can easily delineate the GWPZ [8,15–19].

Different methods have been used for the delineation of GWPZ and their mapping. Some of the research works used multi-criteria decision making (MCDM) [17,20–22], logistic regression [23–26], evidential belief function [25,27], artificial neural network model [28], machine-learning techniques, such as random forest and maximum entropy [29], and many others. Recently, a combination of MCDM and GIS techniques was found as most popular for resolving different complex decision-making problems in natural resources, including investigation of groundwater potential zones [30–38]. Among several MCDM techniques, such as fuzzy set theory (FST) [39–41], data envelopment analysis (DEA) [42], TOPSIS [43,44], ELECTRE [45], multi influencing factor (MIF) [46], and PROMETHEE [47], the analytic hierarchy process (AHP) is widely used and much more common for the groundwater potential zones delineation, especially because of the possibility for integration in GIS [48–52]. This method was developed by Professor Thomas L. Saaty in 1977 and received most attention in natural resource management applications [53]. It is based on generating the weight for each criterion according to the decision maker's pairwise comparisons of the criteria. These weights are crisp numbers that do not include any uncertainty associated with weights. In order to overcome this problem, fuzzified AHP technique was applied in this study. This fuzzy AHP method was developed by VAN Laarhoven and Pedrycz in 1983 [54] and uses fuzzy numbers instead of crisp numbers for solving multi-criteria decision problems. Since then, this method has been applied as solution for different decision-making problems [39,55–61].

Considering that, we used a combination of GIS and FAHP techniques to delineate GWPZ map examining the case of municipality where people are heavily dependent on groundwater resources for their domestic, agricultural, and industrial requirements. We examined the pilot work on the municipality to see if it is possible to apply this method for upscaling on the whole agriculture region (Vojvodina Province) where geomorphological,

geological, and soil conditions are very similar. Considering that, if this methodology gives successful results on this pilot area, we will prove that it could be useful to also apply it to the wider area. Besides, we tried to show that even in cases where there are no available field measurement data, the whole process generating effective results for decision makers could be successfully performed using a small data set created by only collecting physical geographical data. For that purpose, six different decision criteria (geomorphology, geology, soil, slope, drainage density, and land use/land cover) relevant to groundwater storage were used for delineating the groundwater potential zones using FAHP as the most common multi-criteria method in water resource management.

## 2. Materials and Methods

### 2.1. Study Area

The Municipality of Titel is geographically situated at the confluence of the Danube and Tisa rivers, in the Vojvodina region (Republic of Serbia). It lies between latitudes 45°08′02″ N and 45°21′31″ N and longitudes 20°03′09″ E and 20°18′50″ E and covering a total area of 261 km$^2$ (Figure 1). The topography of the study area ranges from 70 m AMSL in the alluvial plain to 126 m AMSL at the loess plateau. The climate of the territory of Vojvodina is mainly controlled by the geographical position in the southern part of the Pannonian Basin. It is moderately continental due to the weaker impact of western air currents and the greater impact of Eurasian continental climate conditions. Winter seasons are cold (January is the coldest month; the average monthly temperatures range from <0.0 °C to 1.0 °C), while summers are hot and humid (July is the warmest month, with an average monthly temperature of between 21.0 °C and 23.0 °C), with a huge range of extreme temperatures (the difference between the highest and lowest temperatures reaches ~70 °C) and very irregular distribution of rainfall per month (an extremely rainy period can be identified in early summer (June) and in periods with low precipitation—November and March) [62]. Climate is also influenced by surface wind, which blows from two prevailing opposite directions; from NW when it is cold and humid, and from SE when it is warm and dry. Rainfall regime is reflected in the pronounced variability in both space and time. The average annual precipitation is 606 mm, with the highest amounts in June and lowest in February [63,64]. During the summer season, the total monthly amount of precipitation can fall within a single day. The lowest average annual rainfall amount of about 540 mm is recorded in the north of the province, while the highest average precipitation values are recorded in the southwest of Vojvodina. The above-mentioned range of temperature values and nonequal distribution of monthly rainfall influence the presence of different values of aridity types [65–67]. As shown by Hrnjak et al. [66], the last two thirty-year cycles of precipitation and air temperature data indicate that the climate of north Serbia is mostly characterized as semi-humid and humid type. De Marton [68], in his approach, used the value of aridity index (IDM) of 28 to perform climatic classification of the given area by distinguishing semi-humid from humid climate. For north Serbia (and the case study area as well), it has been shown that ~75% of the territory is characterized by humid climate. Based on Köppen's climate classification, it can also be pointed out that the investigated area belongs to the humid continental climate with a mean annual temperature of 11.6 °C and average annual rainfall of approximately 625 mm. In the study area, the major soil types are Calcic Chernozem (Glossic) and Mollic Gleysol (Clayish). The geomorphology of the terrain is very diverse, where the main categories of landforms are the alluvial plain, loess plateau, higher river terrace, and lower river terrace. According to land use classification, agricultural land is most common on this territory and because of that, groundwater has a significant role in the municipality.

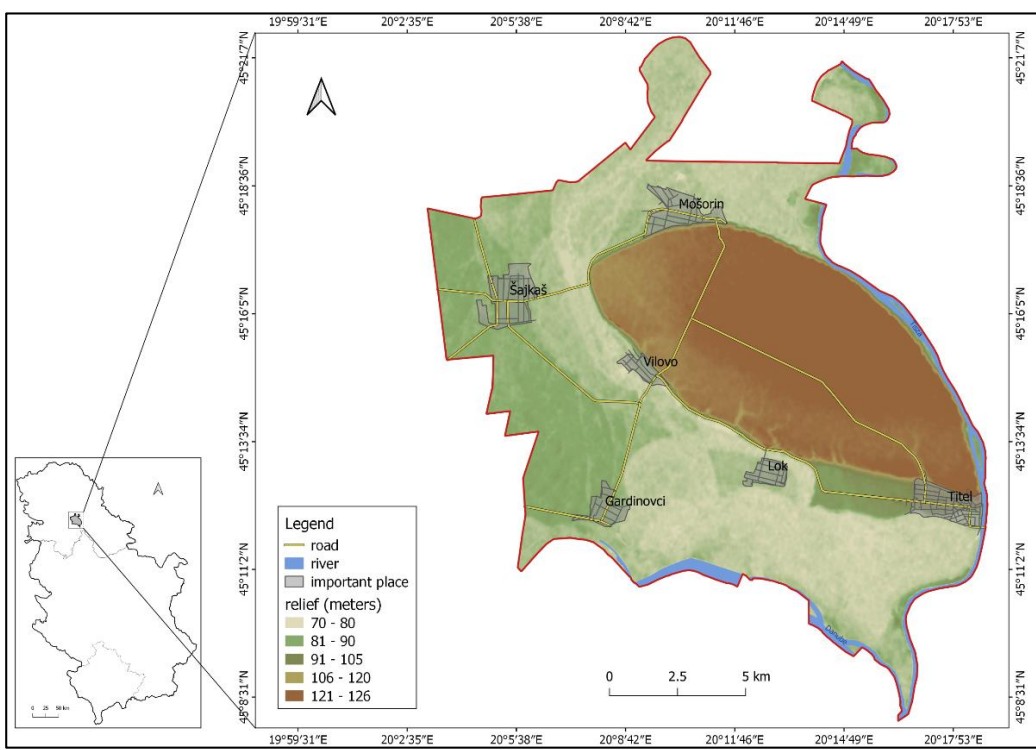

**Figure 1.** Study area—the Titel Municipality.

When it comes to the hydrology of the study area, there are two main rivers—Tisza (164 km in Vojvodina) and Danube (288 km in Vojvodina). Hydrogeological characteristics of the territory have conditioned the connection between surface hydrography and groundwater. These rivers highly affect groundwater because there is intensive water exchange between them. Considering the climatic condition described above, this study area belongs to the climatic type of water regime of phreatic aquifer determined based on the dominant influence of the climatic factor on its formation, primarily precipitation and temperature conditions. In addition to the climatic condition, we can single out the hydrological-anthropogenic type of the water regime of the phreatic issue [69]. The reason for that is the hydro-technical and melioration works conducted during the 18th, 19th, and 20th centuries.

One part of the works was to construct 600 km of the hydro system Danube—Tisza—Danube (HS DTD) network, which began in mid-1947, with the aim to dry up the muddy terrain, increase arable land, and allow navigation on inland waterways. The territory of the Titel Municipality is one of the most endangered because it is located on the lowest landform unit, the alluvial planes of the Danube and Tisa rivers. The problem of inland excess waters led to the very early application of reclamation drainage on these surfaces. In order to protect and transform coastal areas into agricultural land, drainage channels were dug within the detailed channel network HS DTD for the purpose of draining inland excess water. This is very important for groundwater zones because these drainage channels have a big influence on the regime of groundwater in the study area [70,71].

### 2.2. Data Aquisation and Integration into a GIS

In this research paper, different geospatial techniques were applied to delineate the GWPZ in the Titel Municipality. Considering previous research and the availability of the data for this study area, we chose a total of six databases, such as geology, geomorphology, soil, land use/land cover (LULC), slope, and drainage density. All data had to be prepared in GIS environment, using open-source QGIS software.

The identification of GWPZ was performed by the preparation of several maps using different data sources. The base map was prepared using a topographic map (Zrenjanin

sheet, scale 1:100,000) and ASTER DEM (30 m). A geological map was updated using the basic geological map of Serbia (Inđija and Zrenjanin sheets, scale 1:100,000). To obtain geomorphological data, the digitalization of the analog geomorphological map of the autonomous province of Vojvodina, scale 1:300,000, was performed [72]. For digital soil map preparation, we digitized the analog soil map of Vojvodina, scale 1:50,000 [73]. The CORINE Land Cover 2018 was used for the preparation of the LULC map. The ASTER DEM (30 m) data were useful for deriving a slope map presented in degrees using the SLOPE function in QGIS. The drainage density was also extracted from ASTER DEM (30 m) using line density in spatial analyst tools in GIS software.

### 2.3. Methods

In order to delineate the groundwater potential zones of the study area, several thematic maps were prepared as described in previous section. The fuzzy AHP was used to evaluate thematic maps and their features according to their importance for groundwater occurrence. While all steps are summarized in Figure 2, in the next section, the methodology will be described in detail.

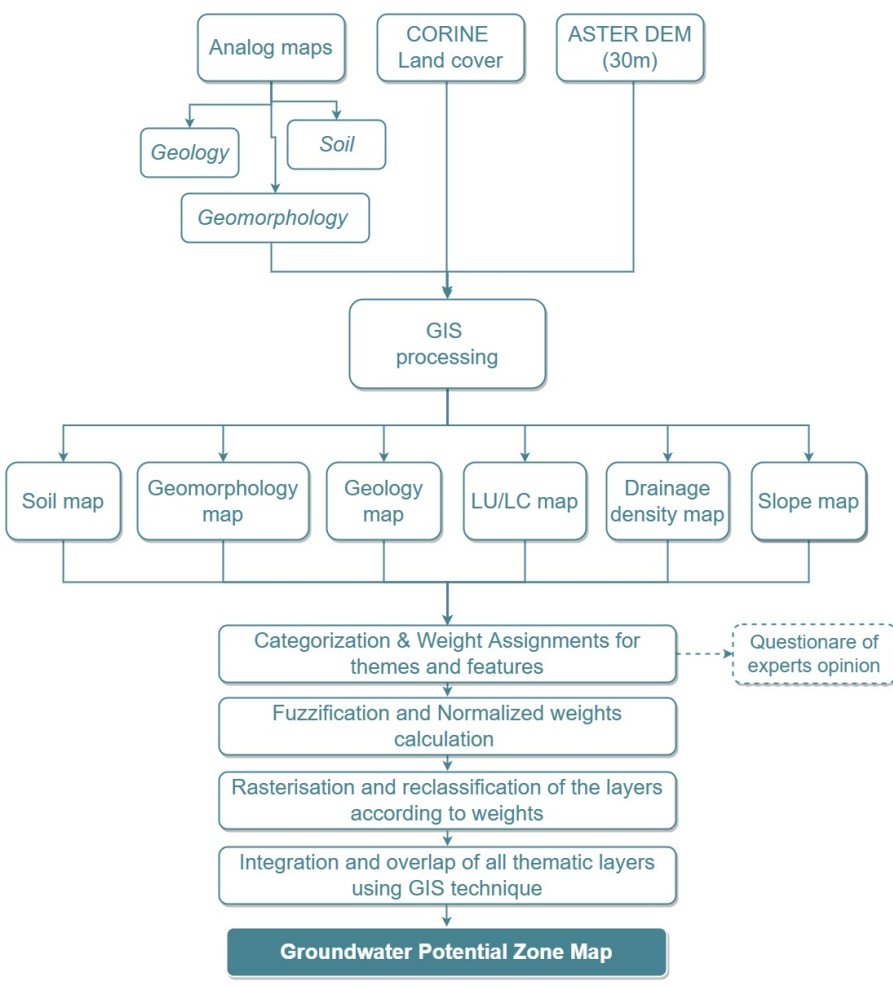

**Figure 2.** Methodological flow chart used in this study.

### 2.3.1. Fuzzy AHP

For the decision-making process, we used AHP developed by Saaty [74]. It is based on the pairwise comparison generated from the experts' judgment between the chosen criteria. Hydrologists', hydrogeologists', environmental and geoscientists', and agriculture experts' opinions were sought to assess the priorities between the most important themes and their features through a questionnaire. A total of six different thematic layers were considered

for this study (Figure 3). The following steps were implemented to determine the weights. These steps were repeated for each feature and theme.

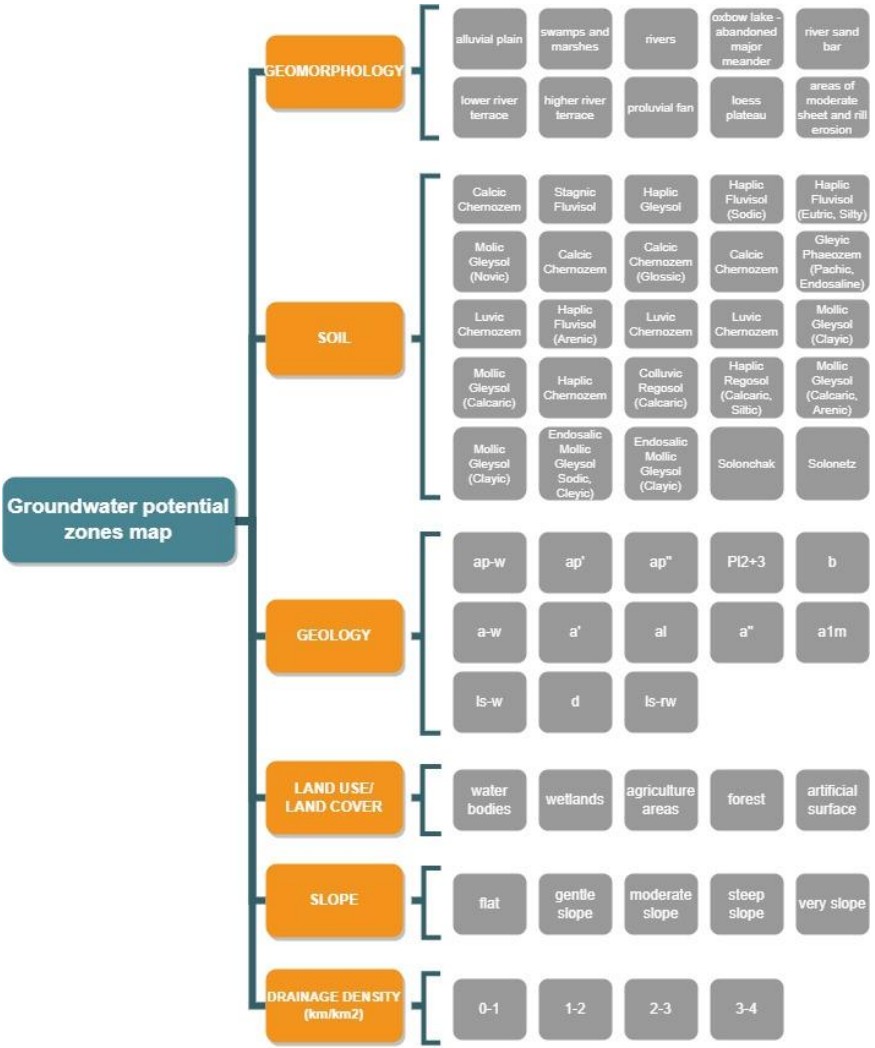

**Figure 3.** Hierarchical tree.

Step I: In order to compare the criteria through a questionnaire, the experts used Saaty's scale (1–9) (Table 1) of relative importance for assigning the weights. A parameter with a low weight illustrates a small impact, and a parameter with a high weight illustrates a high impact on the groundwater potential. Based on the experts' response, the pairwise comparison will be analyzed to establish a judgment matrix.

**Table 1.** Fuzzified Saaty's scale [75,76].

| Importance | Weights | Fuzzy AHP Weights |
|---|---|---|
| Absolutely Important | 9 | (9, 9, 9) |
| Strongly Important | 7 | (6, 7, 8) |
| Fairly Important | 5 | (4, 5, 6) |
| Weakly Important | 3 | (2, 3, 4) |
| Equally Important | 1 | (1, 1, 1) |
| Intermediate Values | 2, 4, 6, 8 | |

Step II: As recommended by Saaty [74], data validation was conducted by calculating the consistency index (CI) and consistency ratio (CR) of the pairwise matrix. For calculating CR, the values of random consistency index (RCI) were obtained from the Saaty's stan-

dard [74] (Table 2). The value of CR should be less than 0.10; otherwise, the corresponding weights have to be re-evaluated to avoid inconsistency.

$$\lambda_{max} = \frac{1}{n} \sum_{i=1}^{n} \frac{\left(\widetilde{M}_w\right)_i}{w_i} \tag{1}$$

$$CI = \frac{(\lambda_{max} - n)}{(n - 1)} \tag{2}$$

$$CR = \frac{CI}{RCI} \tag{3}$$

where $\lambda_{max}$ is the maximum eigenvalue of the judgment matrix $n \times n$, $n$ is the number of themes used in the analysis, $CI$ consistency index, $CR$ consistency ratio, and $RCI$ random consistency index.

**Table 2.** Saaty's ratio index.

|  | 1 | 2 | 3 | 4 | 5 | 6 | 7 | 8 | 9 | 10 | 11 | 12 |
|---|---|---|---|---|---|---|---|---|---|---|---|---|
| RCI value | 0.00 | 0.00 | 0.58 | 0.90 | 1.12 | 1.24 | 1.32 | 1.41 | 1.45 | 1.49 | 1.51 | 1.48 |

Once the appropriate CRs were achieved (Table 3), the next step could be followed.

**Table 3.** The number of themes and features (*n*), the largest eigenvalue of pairwise comparison judgment matrix ($\lambda\_max$), consistency index (*CI*), random consistency index (*RI*), and consistency ratio (*CR*) for the selected criteria to delineate the GWPZ.

| Theme | *n* | $\lambda_{max}$ | *CI* | *RI* | *CR* |
|---|---|---|---|---|---|
| All | 6 | 6.227 | 0.045 | 1.24 | 0.04 |
| Geomorphology | 10 | 10.390 | 0.043 | 1.49 | 0.03 |
| Soil | 25 | 25.710 | 0.030 | 1.66 | 0.02 |
| Geology | 13 | 13.215 | 0.018 | 1.56 | 0.01 |
| LULC | 5 | 5.108 | 0.027 | 1.12 | 0.02 |
| Slope | 3 | 3.12 | 0.065 | 0.58 | 0.11 |
| Drainage | 4 | 4.010 | 0.003 | 0.90 | 0.004 |

Step III: In this step, the crisp numbers in the pairwise comparison matrix were replaced with triangular fuzzy numbers in order to establish the triangular fuzzy judgment matrix [61] (Tables 4 and S1–S6).

$$\widetilde{M} = \begin{bmatrix} (1,1,1) & (a_{12}b_{12}c_{12}) & (a_{13}b_{13}c_{13}) \\ \left(\frac{1}{c_{12}}, \frac{1}{b_{12}}, \frac{1}{a_{12}}\right) & (1,1,1) & (a_{23}b_{23}c_{23}) \\ \left(\frac{1}{c_{13}}, \frac{1}{b_{13}}, \frac{1}{a_{13}}\right) & \left(\frac{1}{c_{23}}, \frac{1}{b_{23}}, \frac{1}{a_{23}}\right) & (1,1,1) \end{bmatrix} \tag{4}$$

where $\widetilde{M}$ present fuzzified pairwise comparison matrix.

**Table 4.** Pairwise comparison matrix for six themes.

|  | Geomorphology | Soil | Geology | LULC | Slope | Drainage Density |
|---|---|---|---|---|---|---|
| Geomorphology | 1/1/1 | 1/2/3 | 1/1/1 | 2/3/4 | 3/4/5 | 4/5/6 |
| Soil | 1/3,1/2,1 | 1/1/1 | 1/2/3 | 2/3/4 | 3/4/5 | 3/4/5 |
| Geology | 1/1/1 | 1/3,1/2,1 | 1/1/1 | 2/3/4 | 3/4/5 | 3/4/5 |
| LULC | 1/4,1/3,1/2 | 1/4,1/3,1/2 | 1/4,1/3,1/2 | 1/1/1 | 1/2/3 | 2/3/4 |
| Slope | 1/5,1/4,1/3 | 1/5,1/4,1/3 | 1/5,1/4,1/3 | 1/3,1/2,1 | 1/1/1 | 1/2/3 |
| Drainage density | 1/6,1/5,1/4 | 1/5,1/4,1/3 | 1/5,1/4,1/3 | 1/4,1/3,1/2 | 1/3,1/2,1 | 1/1/1 |

Step IV: To estimate the fuzzy geometric mean and fuzzy weights of each criterion, Buckely's geometric mean method was applied [57]:

$$\widetilde{r}_i = (\widetilde{a}_{i1} \otimes \widetilde{a}_{i2} \otimes \ldots \otimes \widetilde{a}_{in})^{1/n} \tag{5}$$

$$\widetilde{w}_i = \widetilde{r}_i \otimes (\widetilde{r}_i \otimes \ldots \otimes \widetilde{r}_i)^{-1} \tag{6}$$

where $\widetilde{a}_{in}$ present the fuzzy comparison value of criterion $i$ to criterion $n$ where $\widetilde{r}_i$ is geometric mean of fuzzy comparison rate of criterion $i$ to all; $\widetilde{w}_i$ is the fuzzy weight of the $i_{th}$ criterion.

Step V: At the end, defuzzification, which means changing a fuzzy number ($l$—low possible value, $m$—the most likely value, $u$—the highest possible value) into a crisp number, was performed by calculating the arithmetic mean and then the normalized weight.

$$w_i = (l + m + u)/3 \tag{7}$$

$$NW = \sum_{j=1}^{n} \frac{w_{i1}}{\sum_{i=1}^{n} w_i} + \frac{w_{i2}}{\sum_{i=1}^{n} w_i} + \ldots + \frac{w_{in}}{\sum_{i=1}^{n} w_i} = 1 \tag{8}$$

where $w_i$ is weight, and $NW$ is normalized weight of each theme and feature.

As fuzzification of the pairwise matrices occurred after the experts assigned the weight and pairwise matrices created, firstly, we finished the whole process with the crisp number of Saaty scale. After that, we went back and fuzzified the pairwise matrices, which were compared with previous results. Comparing the results, we noticed that the normalized weights of traditional AHP and FAHP technique, using this fuzzified AHP scale (Table 1), are negligible (Table S7). Considering that, as well as the efficiency in dealing with uncertainties among decision makers' assignment of weight using crisp numbers, we continued our research by applying FAHP as a newer and expanded version of AHP.

### 2.3.2. Classification of the Thematic Maps

After the process of weight assignment was complete, all thematic layers and their features were rasterized and reclassified based on the fuzzy weights. With this, the raster data were prepared for calculating groundwater potential index (GWPI). The whole process was conducted using QGIS software.

### 2.3.3. Delineation of Groundwater Potential Zones

To generate the GWPZ map, all thematic layers and features with relative importance were used. The GIS approach uses the raster data for overlapping, where each pixel of each theme has the same geolocation. In that way, it is possible to generate one output layer integrating characteristics of several layers. Following these steps, GWPI was calculated as output using the following equation from Malczewski et al. [77]:

$$GWPI = \sum_{i=1}^{a} \sum_{j=1}^{b} (W_i \times X_j) \tag{9}$$

where, $GWPI$ —groundwater potential index, $W_i$ —weight of the ith theme, $X_j$ —weight of the jth features, $a$ —the total number of themes, and $b$ —the total number of features in a theme. According to GWPI, the final GWPZ map was classified as very poor, poor, moderate, good, and very good.

### 3. Results and Discussion

In this section, we are going to present the results and discuss the various thematic maps and features, their weights assigned through FAHP, and the final GWPZ map.

### 3.1. Weight Normalization for Thematic Maps

In this research, the fuzzy AHP method was applied for potential groundwater zone delineation. This technique was used for assigning the weight to each theme and their features (Table 5) according to expert opinion and knowledge. The highest weight was given to geomorphology, followed by soil, geology, land use/land cover, slope, and drainage density. Through several iterations, appropriate weights were assigned, and adequate consistency ratios were gained. Fuzzy weights were assigned to each raster file, and further analyses were conducted in QGIS environment.

**Table 5.** Fuzzy AHP weights for six themes and their features for groundwater potential zones delineation.

| Theme | Feature | Feature Weights ($X_j$) | Theme Weights ($W_i$) |
|---|---|---|---|
| Geomorphology | Swamps and marshes | 0.170 | 0.292 |
| | River | 0.157 | |
| | Alluvial plain | 0.184 | |
| | Oxbow—lake—abandoned major meander | 0.120 | |
| | River sand bar | 0.100 | |
| | Lower river terrace | 0.090 | |
| | Higher river terrace | 0.065 | |
| | Areas of moderate sheet and rill erosion | 0.037 | |
| | Proluvial fan | 0.041 | |
| | Loess plateau | 0.036 | |
| Soil | Stagnic Fluvisol | 0.067 | 0.264 |
| | Calcic Chernozem | 0.067 | |
| | Haplic Gleysol | 0.065 | |
| | Haplic Fluvisol (Sodic) | 0.067 | |
| | Haplic Fluvisol (Eutric, Silty) | 0.056 | |
| | Molic Gleysol (Novic) | 0.056 | |
| | Calcic Chernozem | 0.056 | |
| | Calcic Chernozem (Glossic) | 0.056 | |
| | Calcic Chernozem | 0.056 | |
| | Gleyic Phaeozem (Pachic, Endosaline) | 0.056 | |
| | Luvic Chernozem | 0.038 | |
| | Haplic Fluvisol (Arenic) | 0.038 | |
| | Luvic Chernozem | 0.038 | |
| | Luvic Chernozem | 0.045 | |
| | Mollic Gleysol (Clayic) | 0.035 | |
| | Mollic Gleysol (Calcaric) | 0.026 | |
| | Colluvic Regosol (Calcaric) | 0.022 | |
| | Haplic Chernozem | 0.022 | |
| | Mollic Gleysol (Calcaric, Arenic) | 0.022 | |
| | Mollic Gleysol (Clayic) | 0.022 | |
| | Endosalic Mollic Gleysol (Clayic) | 0.021 | |
| | Endosalic Mollic Gleysol Sodic, Cleyic) | 0.022 | |
| | Solonchak | 0.018 | |
| | Haplic Regosol (Calcaric, Siltic) | 0.018 | |
| | Solonetz | 0.011 | |
| Geology | ap-w—Flood sediments of the second fluvial terrace | 0.123 | 0.212 |
| | ap'—Flood sediments of the first fluvial terrace | 0.131 | |
| | ap"— Flood sediments | 0.133 | |
| | $Pl_{2+3}$—Gravel, siltstone sand, sandy-clayey siltstones | 0.074 | |
| | b—Sediments of swamps | 0.074 | |
| | a–w—Stream bed sediments of the second fluvial terrace | 0.072 | |
| | a'—Stream bed sediments of the first fluvial terrace | 0.074 | |
| | Al—Alluvium | 0.074 | |
| | a"— Stream bed sediments | 0.074 | |
| | $a_1m$—Clay and siltstone in oxbow lake | 0.074 | |
| | ls-w—Lesoidal on the surface of the second fluvial terrace | 0.048 | |
| | d—Deluvium | 0.016 | |
| | ls-rw—Lesoidal on the loess plateau | 0.032 | |
| LULC | Water bodies | 0.419 | 0.109 |
| | Wetlands | 0.285 | |
| | Agriculture areas | 0.179 | |
| | Forest | 0.080 | |
| | Artificial surface | 0.037 | |

**Table 5.** *Cont.*

| Theme | Feature | Feature Weights ($X_j$) | Theme Weights ($W_i$) |
|---|---|---|---|
| Slope (degrees) | Flat (0–5) | 0.730 | 0.073 |
| | Moderate (5–11) | 0.21 | |
| | Steep (11–31) | 0.06 | |
| Drainage density (km/km$^2$) | Very low (0–1) | 0.496 | 0.051 |
| | Low (1–2) | 0.284 | |
| | Moderate (2–3) | 0.143 | |
| | Very high (3–4) | 0.077 | |

*3.2. Thematic Layer Analysis*

3.2.1. Geomorphology

The geomorphology of an area is significant information used for the delineation of GWPZ, which controls the subsurface movement of groundwater [78]. In this study, it has a weight of ($W_i$) 29.21%, ranked as the most important criterion for GWPZ delineation. Knowledge about geomorphology is crucial for the development and effective management of groundwater resources of an area [16]. In the landscape of the Titel Municipality, we noticed about thirteen types of landforms. The main categories of landforms are the alluvial plain, which covers about 38% (98.60 km$^2$) of the area, then loess plateau (30%, 77.26 km$^2$), higher river terrace (15%, 38.91 km$^2$), and lower river terrace (11%, 28.35 km$^2$). Other forms have a much smaller area, and they are areas of moderate sheet and rill erosion (2.9%, 7.58 km$^2$), proluvial fan (0.78%, 2.03 km$^2$), swamps and marshes (0.27%, 0.71 km$^2$), river sand bar (0.12%, 0.31 river sand bar), and oxbow-lake-abandoned major meander (0.04%, 0.11 km$^2$). According to previous research, we recognized that alluvial plain, oxbow lake, river sand bar, and swamps and marshes have a big potential for groundwater storage. On the other hand, loess plateau with cracks provides a system for rapid water transmission to the deepest layers, so it has the smallest significance related to collecting, infiltrating, and occurring of groundwater. The geomorphological map is shown in Figure 4.

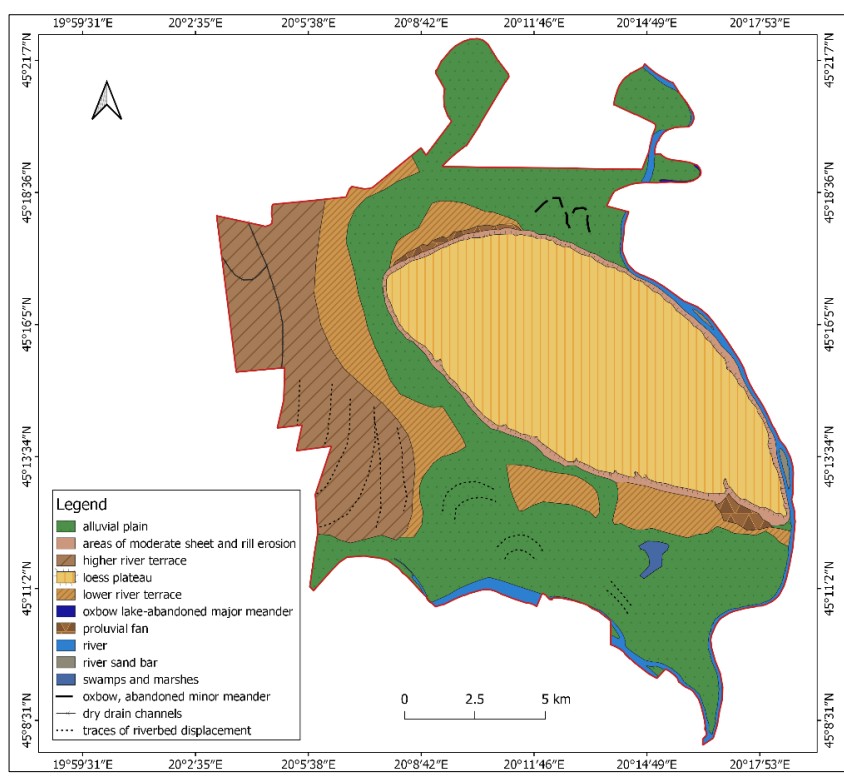

**Figure 4.** Main geomorphological units/relief units of the study area.

### 3.2.2. Soil

Soil characteristics have a considerable role in the infiltration of water, and it occurs second with 26.35% ($W_i$). The rate of infiltration largely depends on the soil texture and related hydraulic characteristics of the soils [13,16]. According to World Reference Base for Soil Resources [79], seven soil classes, such as Fluvisol, Chernozem, Regosol, Gleysol, Solonchak, Phaeozem, and Solonetz, are distributed across the study area (Figure 5). They have a different variety according to their physical, water-physical, mechanical, and chemical properties (Table 6).

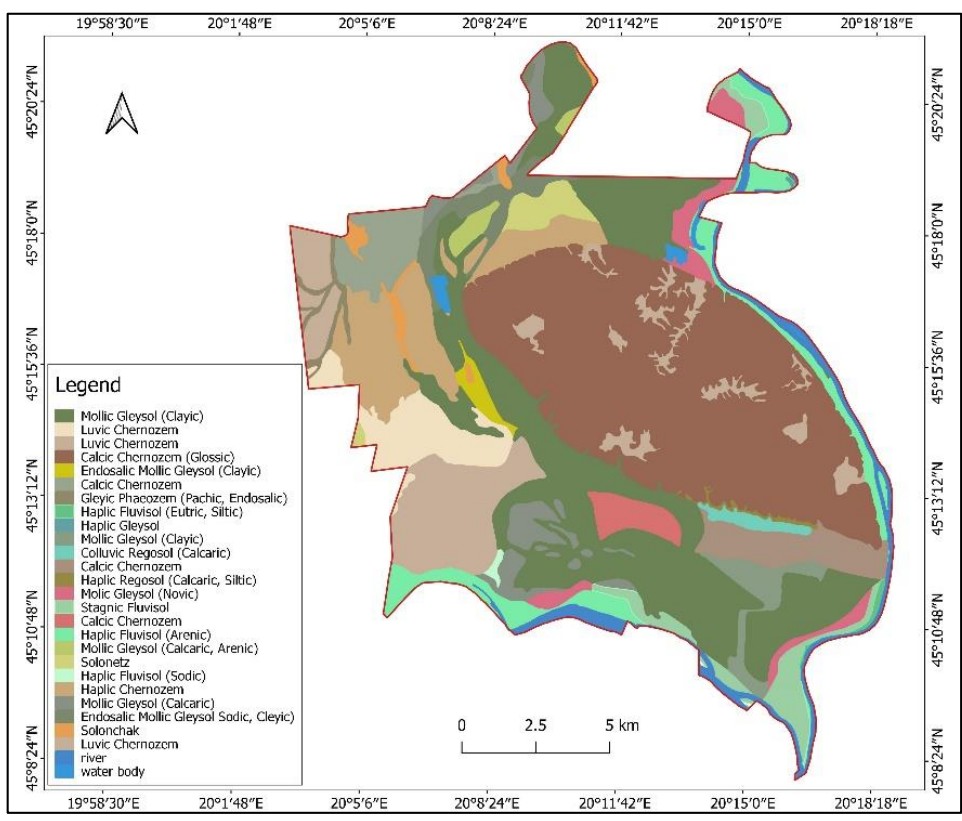

**Figure 5.** Main soil types of the study area.

Calcic Chernozem, Stagnic Fluvisol, Molic Gleysol (Novic), Haplic Gleysol, different varieties of Fluvisol, have a good water-holding capacity. This type of soil is distributed in the area of loess plateau, higher and lower river terrace, and in the alluvial plain. Soils such as Solonetz, Solonchak, different varieties of Regosol, and different varieties of Gleysol (except Molic Gleysol (Novic) and Haplic Gleysol) have poor water-holding capacity and were ranked significantly lower. These are mostly located in the lower parts of the study area with small distribution on lower and higher river terraces.

**Table 6.** Soil type over the research area and its area expressed in km$^2$ and percentage.

| Soil Type | Area (km$^2$) | Area (%) |
|---|---|---|
| Calcic Chernozem (Glossic) | 73.17 | 29.09 |
| Mollic Gleysol (Clayic) | 53.69 | 21.34 |
| Haplic Chernozem | 16.09 | 6.40 |
| Haplic Fluvisol (Arenic) | 11.97 | 4.76 |
| Luvic Chernozem | 14.02 | 5.57 |
| Luvic Chernozem | 12.18 | 4.84 |
| Luvic Chernozem | 9.32 | 3.71 |
| Stagnic Fluvisol | 7.20 | 2.86 |

**Table 6.** *Cont.*

| Soil Type | Area (km$^2$) | Area (%) |
|---|---|---|
| Mollic Gleysol (Calcaric) | 7.8 | 3.10 |
| Calcic Chernozem | 7.63 | 3.03 |
| Calcic Chernozem | 6.39 | 2.54 |
| Molic Gleysol (Novic) | 5.36 | 2.13 |
| Endosalic Mollic Gleysol Sodic, Cleyic) | 5.43 | 2.16 |
| Mollic Gleysol (Clayic) | 4.33 | 1.72 |
| Calcic Chernozem | 3.42 | 1.36 |
| Solonetz | 3.21 | 1.28 |
| Solonchak | 2.52 | 1 |
| Mollic Gleysol (Calcaric, Arenic) | 1.66 | 0.66 |
| Endosalic Mollic Gleysol (Clayic) | 1.54 | 0.61 |
| Gleyic Phaeozem (Pachic, Endosalic) | 1.5 | 0.60 |
| Colluvic Regosol (Calcaric) | 1.34 | 0.53 |
| Haplic Regosol (Calcaric, Siltic) | 0.93 | 0.37 |
| Haplic Fluvisol (Eutric, Siltic) | 0.51 | 0.20 |
| Haplic Fluvisol (Sodic) | 0.3 | 0.12 |
| Haplic Gleysol | 0.04 | 0.02 |

### 3.2.3. Geology

The weight ($W_i$) of geology classes is 21.18%, ranking third among the criteria selected for GWPZ mapping in this study area.

The geology of terrain plays a significant role in the occurrence and distribution of groundwater [16,78]. Some studies [80,81] take into account the geological factor because of its strong influence on water percolation, but some researchers, such as Edet et al. [82], neglected this factor because they consider that information about lithology is provided by the lineaments and drainage characters. Quaternary sediments are dominant in the study area. The Pleistocene (ls—w, ap—w, a—w, ls—rw) and the Holocene (al, b, ap″, a, a′, $a_1$m, d) sediments occurring in the form of sandy clay aleurite, aleurite sandy, gravel, aleurite clay, brown clay, and loess are the most widespread. In the south of the study area, sediments of the middle-upper Pliocene ($Pl_{2+3}$) occur. Alluvium has the highest potential for groundwater storage, so all formations were ranked with highest rank ($X_j$). From the opposite side, lesoidal on the loess plateau was ranked with the lowest rank. The spatial distribution of major geological classes in the study area is shown in Figure 6, while the area of each geology unit is presented in Table 7.

**Table 7.** Geology units over the research area and their area expressed in km$^2$ and percentage.

| Category | Geology Units | Area (km$^2$) | Area (%) |
|---|---|---|---|
| ls—rw | Lesoidal on the loess plateau | 77.26 | 29.62 |
| ls—w | Lesoidal on the surface of the second fluvial terrace | 53.60 | 20.55 |
| $a_1$m | Clay and siltstone in oxbow lake | 32.17 | 12.33 |
| a″ | Flood sediments | 27.81 | 10.66 |
| $Pl_{2+3}$ | Gravel, siltstone sand, sandy—clayey siltstones | 15.58 | 5.97 |
| Al | Alluvium | 15.02 | 5.76 |
| a′ | Stream bed sediments of the first fluvial terrace | 11.61 | 4.45 |
| ap—w | Flood sediments of the second fluvial terrace | 8.59 | 3.29 |
| ap′ | Flood sediments of the first fluvial terrace | 7.42 | 2.85 |
| a—w | Stream bed sediments of the second fluvial terrace | 7.14 | 2.74 |
| b | Sediments of swamps | 2.39 | 0.92 |
| d | Deluvium | 1.39 | 0.53 |
| ap″ | Flood sediments | 0.84 | 0.32 |

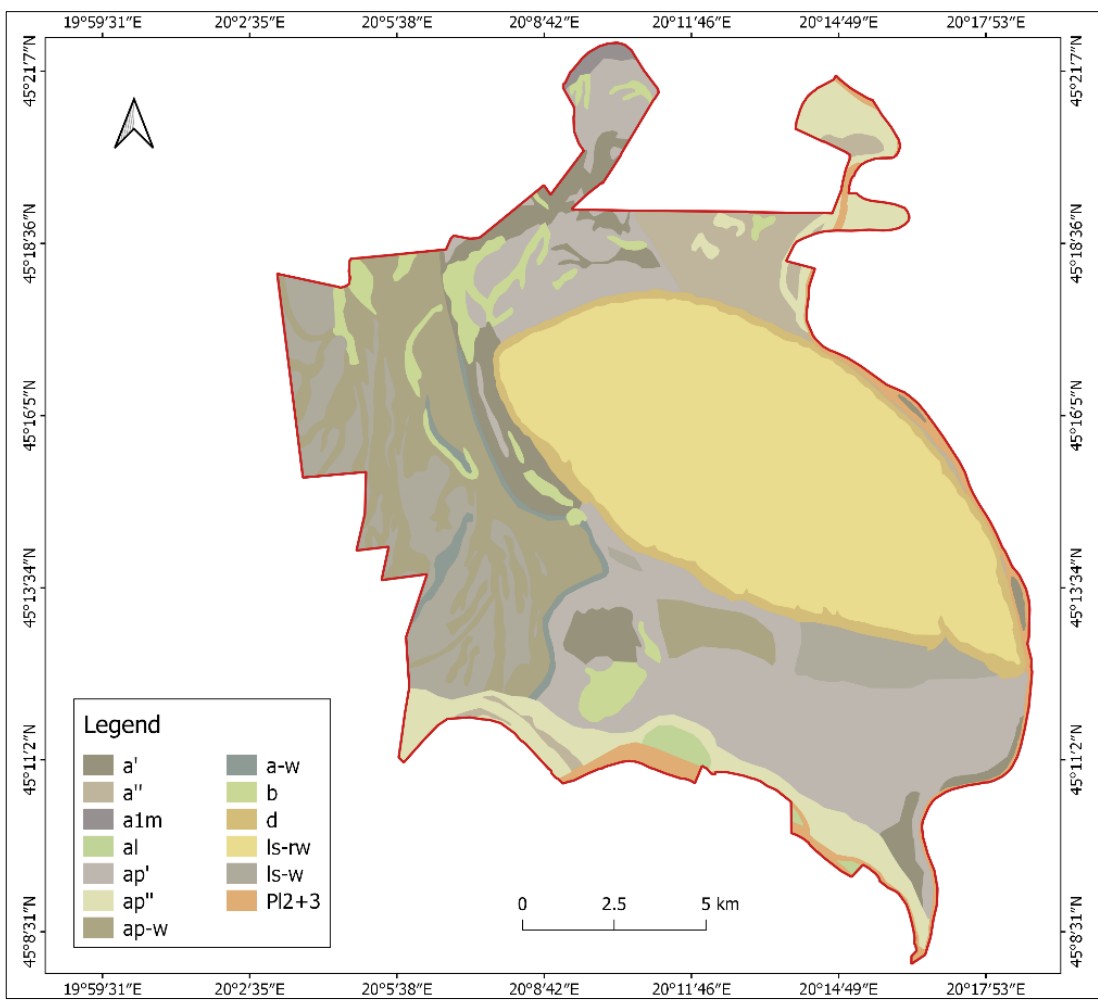

**Figure 6.** Geology map of the study area (ap—w—flood sediments of the second fluvial terrace; ap'—flood sediments of the first fluvial terrace; ap"—flood sediments; $Pl_{2+3}$—gravel, siltstone sand, sandy—clayey siltstones; b—sediments of swamps; a—w-stream bed sediments of the second fluvial terrace; a'—stream bed sediments of the first fluvial terrace; al—Alluvium; a"—stream bed sediments; $a_1m$—clay and siltstone in oxbow lake; ls—w—lesoidal on the surface of the second fluvial terrace; d—deluvium; ls—rw—lesoidal on the loess plateau).

### 3.2.4. Land Use/Land Cover

Land use/land cover ranks fourth, with 10.87% $W_i$, and it is an important factor for the development of groundwater resources [83]. The nature of surface materials is necessary to help quantify the water budget because it has a significant influence on control runoff and infiltration [7,84]. Evapotranspiration, volume, and recharge of the groundwater are also affected by LULC [2]. In the present work, standard visual interpretation methods show several classes that give the essential information on groundwater, soil moisture, infiltration, etc., in addition to providing an indication of groundwater requirements. The LULC map shows that by far the largest area in the municipality is occupied by agricultural land, about 82% (215.17 km$^2$), followed by forest areas (9%, 23.71 km$^2$), artificial surface (5%, 12.91 km$^2$), water bodies (3%, 7.17 km$^2$), and wetlands (1%, 1.86 km$^2$) (Figure 7).

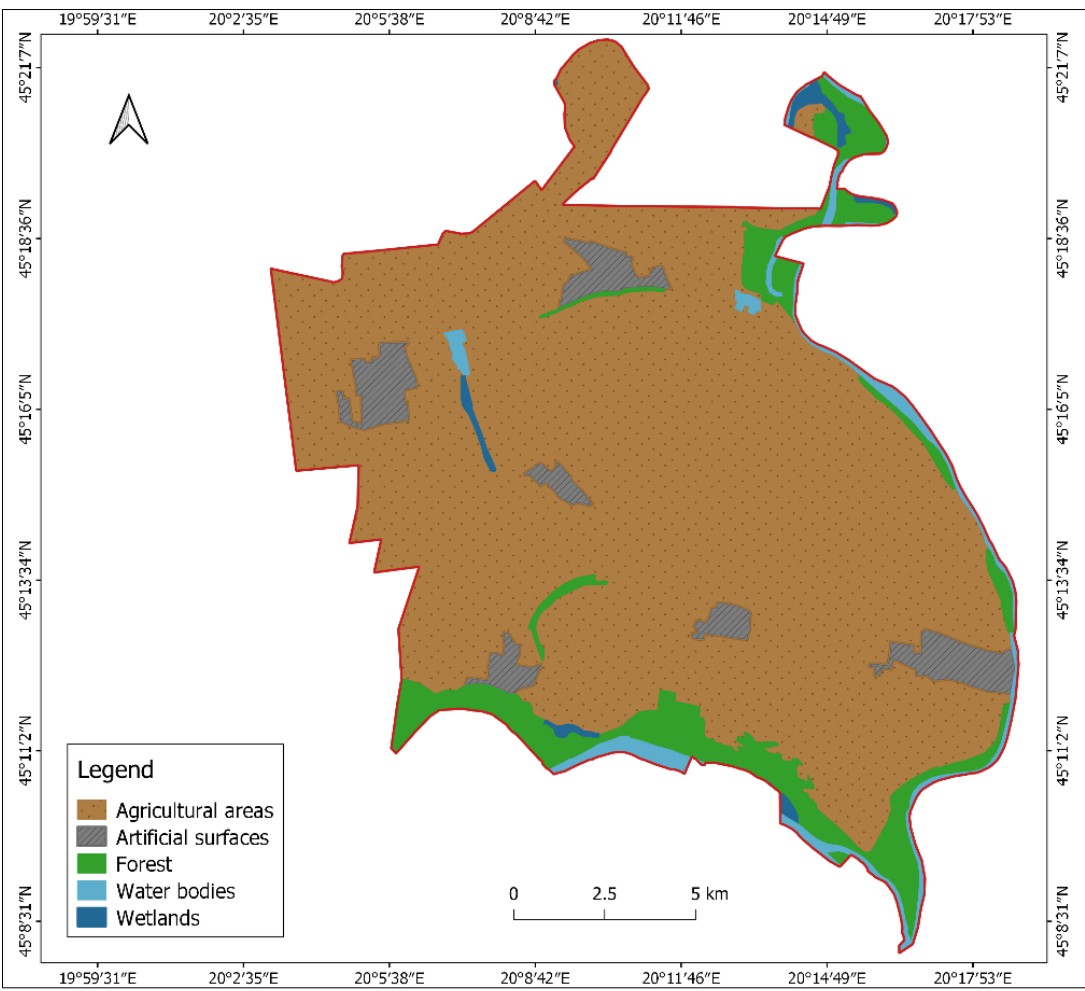

**Figure 7.** Land use/land cover map of the study area.

3.2.5. Slope

Slope takes place in fifth order, with 7.3% $W_i$. A slope is a change of elevation of a surface and the principal factor of the superficial water flow. Slope is a significant terrain characteristic and is directly proportional to the runoff and groundwater recharge. According to Fox et al. [85], infiltration rate decreased until 11.5°, and with further increase of the slope, infiltration remained the same. Considering that, we categorized slope into three classes where slopes larger than 11° were categorized in one class. The three classes are: 0–5° (flat), 5–11° (moderately sloping), and 11–31° (steep sloping). According to Table 8, flat terrain is the most widespread, with 95.81%, while very steeply sloping occupies only 0.04% of the territory. The slope map (in degrees) of the study area is presented in Figure 8.

**Table 8.** Slope classes over the research area and their area expressed in km$^2$ and percentage.

| Slope | Area (km$^2$) | Area (%) |
|---|---|---|
| Flat | 246.77 | 95.81 |
| Moderately sloping | 7.32 | 2.84 |
| Steep sloping | 3.46 | 1.34 |



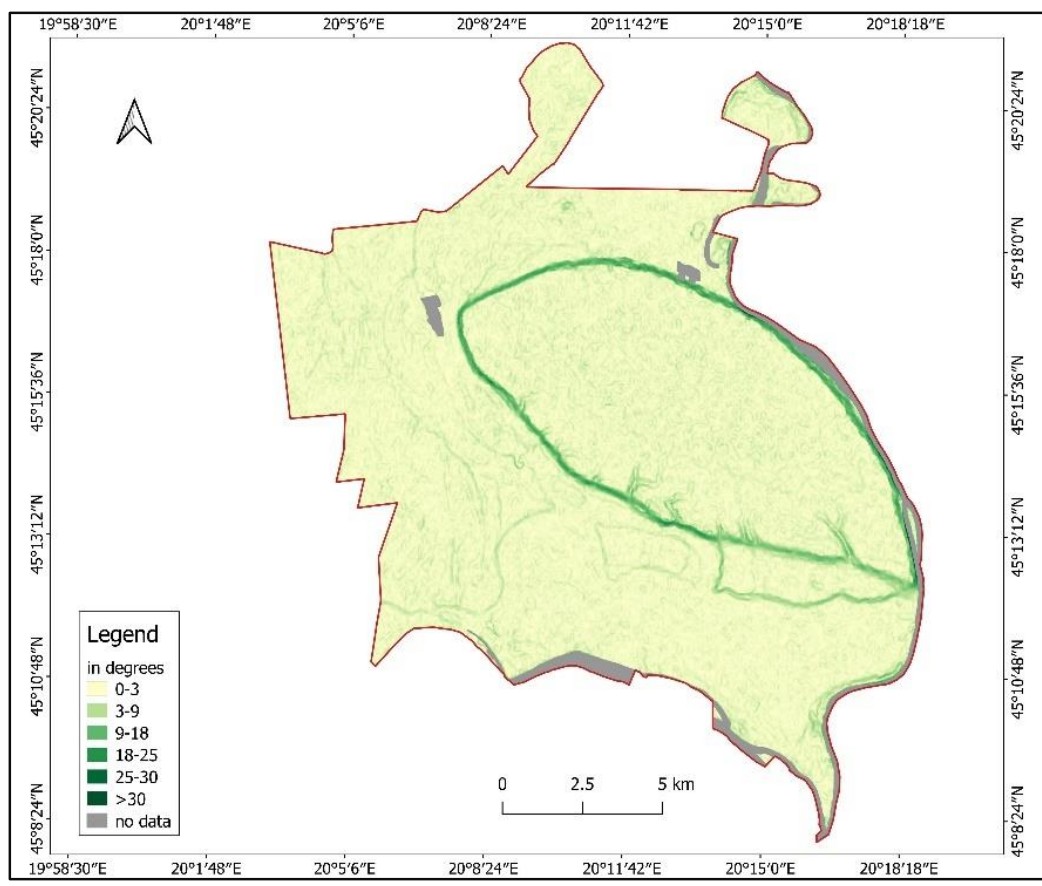

**Figure 8.** Slope map of the study area.

### 3.2.6. Drainage Density

In sixth place is drainage density, with weight of 5.08%. The drainage density (km/km$^2$) is an important component for the assessment of groundwater availability and runoff distribution. The drainage network depends on the lithology, and it provides an important index of infiltration rate. Because of that, it is an indispensable parameter when it comes to the identification of groundwater zones [15,17,86]. In this paper, a drainage density map was extracted using the digital elevation model (DEM). The study area was divided into a grid (cell size 1 km$^2$), and the total length of drains was calculated per cell. The grid size (1 km$^2$) was considered appropriate according to the complexity of the terrain and the area of the municipality. Further, by dividing the total length of all drains in a drainage basin by the total area of the drainage basin [87,88], the map of drainage density was generated. This process was performed in GIS environment following Equation (10):

$$DD = \sum L_{ws}/A_{ws} \qquad (10)$$

where
$DD$—drainage density;
$L_{ws}$—total length of drains in the drainage basin;
$A_{ws}$—area of the drainage basin.

Drainage density was reclassified and categorized as very low (<1 km/km$^2$) when occupying 31.79% (180.9 km$^2$), low (1–2 km/km$^2$) with 47.79% (271.92 km$^2$), moderate (2–3 km/km$^2$) with 17.6% (100.14 km$^2$), and very high (3–4 km/km$^2$) covering 2.82% (16.05 km$^2$). Figure 9 depicts the drainage density map of the Titel Municipality.

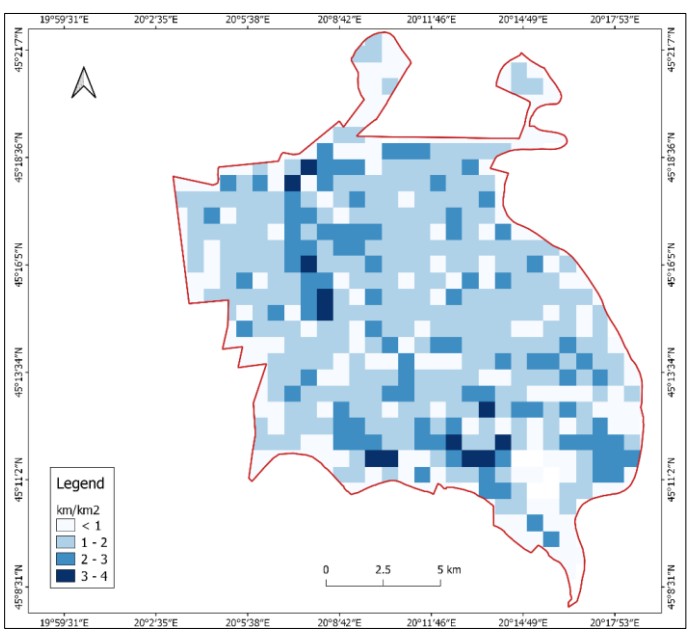

**Figure 9.** Drainage density (km/km$^2$) map of the study area.

### 3.2.7. Groundwater Potential Zone (GWPZ)

Groundwater is a very important renewable resource, but drastic population growth coupled with increasing need for water resources for food production, anthropogenic activities within agricultural production, industrial purposes, and domestic needs, all have a strong influence on discharging of groundwater reservoirs. A better understanding of the groundwater potential is of supreme importance for the planning and sustainable development of water management of an area. Such information is essential for the design and implementation of structures for corrective measures to improve groundwater recharge processes.

GWPZ map (Figure 10) of the Titel Municipality was generated through the integration of six thematic maps. For each thematic layer, the weight was given according to its importance in groundwater occurrence, storage, and movement (Table 3). This judgement by experts was formulated using the FAHP technique. By calculating the GWPI, we obtained values and classified them into five classes (Table 9).

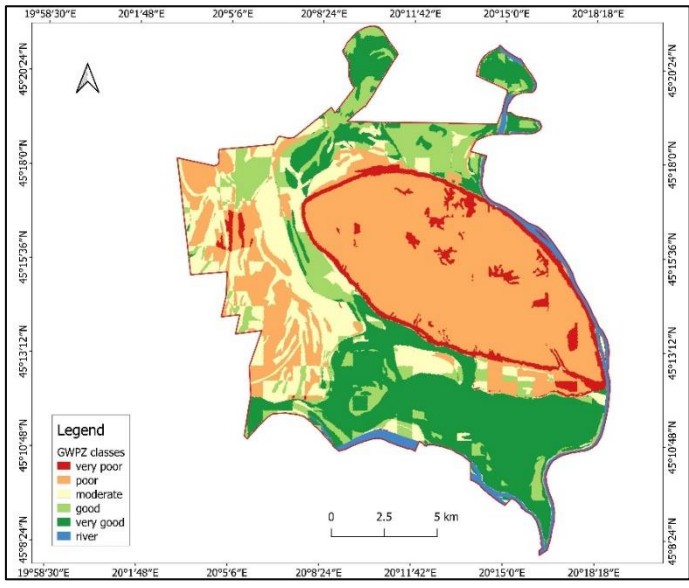

**Figure 10.** Groundwater potential zone classification map of the Titel Municipality.

**Table 9.** GWPZ classes in the Titel Municipality.

| GWPZ | Area (km$^2$) | Area (%) |
|---|---|---|
| Very poor | 14.63 | 5.7 |
| Poor | 106.09 | 41.34 |
| Moderate | 38.95 | 15.18 |
| Good | 31.05 | 12.10 |
| Very good | 65.91 | 25.68 |

The Municipality of Titel has 25.68% (65.91 km$^2$) of the area classified as very good. The whole alluvial plain and north part of the higher river terrace are classified as good and cover 12.10% (31.05 km$^2$) of the study area. Subsequently, 15.18% (38.95 km$^2$) is recognized as moderate, 41.34% (106.09 km$^2$) as poor, and 5.7% (14.63 km$^2$) as very poor, and this zone is identified at the loess plateau and in the artificial areas.

*3.3. Discussion*

Considering the previous research, most of the area in the municipality possesses good and very good groundwater zones (Figure 10). These zones are recognized in the whole alluvial plain of the Danube and the Tisza River, occurring with 77% representation in good and 98% in very good potential areas. The main reason for this is the proximity to the rivers, which has a significant influence on groundwater recharging. Additionally, this is the lowest part of the terrain, which indicates that there is the main "collector" for all groundwater that flows from higher relief units. Besides that, 13% and 10% of the good potential area occur on the lower and higher river terrace, respectively. When it comes to the very poor zone, it is mostly located in the area of moderate sheet and rill erosion, with 46%. Very poor and poor zones are also recognized on the loess plateau, corresponding to 37% and 68% representation in these classes, while the moderate zone occurs on the lower river terrace, with 43%, and higher river terrace, with 36%. The low potential of loess plateau is a consequence of high runoff from this relief unit to the lower parts, as well as poor potential of water retention of lesoidal, which covers the loess plateau.

Infiltration and movement of the groundwater also depend on the soil type, permeability, and porosity. In the study area, several types of the Fluvisol, Gleysol, Chernozem, Regosol, etc., were recognized. As Glaysol is hydric soil, which is seasonal or permanently saturated by groundwater, approximately 70% of the very good and good potential areas of the GWPZ classes are covered by different type of this soil. Besides, 15% is covered by types of Fluvisol, which have good water-holding capacity. Haplic and Luvic Chernozem cover around 51% of the moderate potential zone, while areas of poor and very poor zones are covered approximately 63% by Calcic Chernozem. Even though Chernozem has a good water-holding capacity, the high representation in very poor and poor zones is caused by its location on the loess plateau, which is characterized with very low importance for groundwater occurrence but in the theme with the highest weight.

When it comes to the geology of an area, porosity and permeability of the geological units affect the amount of groundwater storage in sediments [78]. Among the different geological formations in the study area, unconsolidated flood sediments, which spread over alluvial plains, cover around 70% of the very high groundwater potential area. From the opposite side, loess plateau is covered with lesoidal, which is not so permeable and, here, has the poorest potential for groundwater storage.

Land cover type could highly affect the rainwater runoff, intensity of infiltration, as well as intensity of evapotranspiration. It depends on different land cover classes. In this study area, five different land cover classes are recognized. Among them, water bodies and wetlands present a direct source of groundwater recharge [89]. In this area, agriculture land is flat and porous, where runoff is negligible, but infiltration is high, which is a great condition for containing groundwater. The agriculture area is predominantly located in the municipality and largely overlaps with very high (79%) and high (77%) groundwater potential zones. The lowest potential for containing groundwater is presented by the

artificial surface, which could affect surface water runoff, reducing the infiltration from surface to underground.

Surface runoff depends on the degree of slope, which is essential for groundwater recharge. If slopes are larger, then the infiltration will be smaller because the water received from precipitation flows rapidly down a steep slope, and it does not have sufficient residence time to infiltrate to the saturated zone. Otherwise, if it is smaller than the infiltration and recharge, the saturated zone will be larger [12,17,90]. In this study, very good groundwater potential zone is completely below the flat area, while 75% of the good potential zones have a moderate steep slope. Poor potential is recognized in the area of moderate slope (56%) and in the area of steep slope (42%).

The movement and infiltration also depend on the drainage density of some areas [86]. A higher drainage density tends to indicate lower groundwater potential. Low drainage density represents high infiltration and hence contributes more to the groundwater potential. In this study area, 80% of the good and very good groundwater potential area is discovered in the area with low and moderate drainage density.

Generating the results based on systematic, internally consistent, and quantitative evaluation of expert knowledge obtained by the Saaty's scale and extended FAHP multi-criteria analysis, the map of GWPZ was produced for the Municipality of Titel. This pilot project showed that, with this type and amount of data, it is possible to delineate the GWPZ for the area with similar circumstances and where reliable data about groundwater are missing. Nonetheless, this research suffered from the absence of data about groundwater level, as well as meteorological data (precipitation, solar radiation, air temperature). Due to this, we also could not calculate evapotranspiration on high spatial resolution. Further, the hydrodynamic behavior was not also considered in depth, which is another possibility to consider during work on such research. Nevertheless, these six criteria chosen in this study are suitable for the delineation of the GWPZ in moderately continental climate areas. The other climate conditions would require further analysis.

However, even if this methodology is good for solving complex decision-making problems, there are some limitations. Firstly, finding experts in relevant fields for pairwise comparison can be difficult and could be seen as a limitation. Additionally, the number of criteria may be considered a limitation, i.e., the more the number of criteria increases, the more difficult it is to solve a pairwise matrix. In this case, experts need a lot of effort and time to spend on working, and if consistency is not satisfied, the experts have to re-assign the weight [91]. However, no matter which analytical decision tool is chosen, the implementation is complex. In research like this, with limited data, this approach could be used as the most suitable tool for solving different groundwater management problems.

## 4. Conclusions

The present study focused on a probabilistic approach that used a combination of GIS and fuzzy AHP techniques to find the potential groundwater zones in the small Municipality of Titel. Using extended fuzzy AHP methods and expert opinion, a more detailed, systematic, and complex assessment of natural conditions was carried out. A total of six thematic layers, such as geology, geomorphology, LULC, drainage density, soil, and slope, were used to delineate the GWPZ. These thematic maps are integrated and overlaid in QGIS software. Based on it, the GWPZ in the Municipality of Titel were delineated and classified into five classes as very good (25.68%), good (12.10%), moderate (15.18%), poor (41.34%), and very poor (5.7%). These techniques indicated that very good and good groundwater potential zones are predominantly located in the alluvial plains of the Danube and Tisa Rivers, covered with unconsolidated flood sediments. This is also a flat agricultural area with low drainage density, which are good conditions for groundwater storage. Otherwise, very poor and poor potential zones are located in the central part of the study area, on the landform of the loess plateau and in the artificial areas. Combining FAHP and GIS techniques, we found that even with a small and limited data set, this approach could be used for delineation of GWPZ and upscaling on larger region with the

same natural and socio—economic settings. It is confirmed that FAHP presents a good method for complex decision—making problems in the field of groundwater management, generating useful results for decision makers. This is especially important for countries and regions, which abound in groundwater but have poor data and aspire to develop sustainable groundwater management, as in the case of the Vojvodina region. Since most of the area of the municipality is covered by agricultural land, this study will help improve the irrigation facility and develop the agriculture productivity of the area by enhancing groundwater management.

**Supplementary Materials:** The following supporting information can be downloaded at: https://www.mdpi.com/article/10.3390/ijgi11040257/s1, Table S1: The pairwise comparison matrix for geomorphology, Table S2: The pairwise comparison matrix for geology, Table S3: The pairwise comparison matrix for land use/land cover, Table S4: The pairwise comparison matrix for drainage density, Table S5: The pairwise comparison matrix for slope, Table S6: The pairwise comparison matrix for soil, Table S7: The comparison of normalized AHP and normalized FAHP weights.

**Author Contributions:** Conceptualization, Mirjana Radulović and Dragoslav Pavić; methodology, Mirjana Radulović, Sanja Brdar and Dragoslav Pavić; validation, Mirjana Radulović, Sanja Brdar and Dragoslav Pavić; formal analysis, Mirjana Radulović; investigation, Mirjana Radulović; writing—original draft preparation, Mirjana Radulović; writing—review and editing, Mirjana Radulović, Dragoslav Pavić, Minučer Mesaroš, Tin Lukić, Biljana Basarin, Stevan Savić and Sanja Brdar; visualization, Mirjana Radulović; supervision, Dragoslav Pavić and Vladimir Crnojević; funding acquisition, Sanja Brdar and Vladimir Crnojević. All authors have read and agreed to the published version of the manuscript.

**Funding:** The authors acknowledge the financial support of Provincial Secretariat for Higher Education and Scientific Research of Vojvodina through project Development of decision support systems for agricultural production using data fusion and artificial intelligence (Grant No. 142-451-2698/2021-01), the Ministry of Education, Science and Technological Development of the Republic of Serbia (Grants No. 451-03-68/2022-14/200358 and 451-03-68/2022-14/ 200125), and the European Union's Horizon 2020 research and innovation programme under ANTARES project (SGA-CSA. No. 739570 under FPA No. 664387).

**Institutional Review Board Statement:** Not applicable.

**Informed Consent Statement:** Not applicable.

**Data Availability Statement:** The author involved approves the availability of the data obtained from this study. In addition, the GIS input data used for the GWPZs delineation presented in this study are available upon request from the author.

**Conflicts of Interest:** The authors declare no conflict of interest.

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
