# Peer review of "Assessment of Groundwater Potential Zones Using GIS and Fuzzy AHP Techniques—A Case Study of the Titel Municipality (Northern Serbia)"

_ijgi, doi:10.3390/ijgi11040257_

Round 1

Reviewer 1 Report

no comment.

Author Response

Dear reviewer,

Thank you for your support and acceptance of the paper.

Best regards!

Reviewer 2 Report

Accept in present form

Author Response

(The authors gave the same response as above.)

Reviewer 3 Report

Dear Authors,

Thank you for considering and introducing the suggested changes.

The paper presents an interesting research which fits to the scope of the journal, however, I still have some comments which should be taken into consideration before publishing.

I do not understand why a municipality (community) for which there is no basic data was selected? There is also a lack of verification of the obtained results. If there is no available multi-year period data, at least single attempts in the field should be made.

I am still not satisfied with the discussion of results. But thanks to the changes I have noticed an increase in the scientific quality of the article. Perhaps it is worth expanding the introduction and discussion chapter a bit. Below are some examples from Central Europe that may be useful.

  1. Worsa-Kozak M, Zimroz R, Michalak A, Wolkersdorfer C, Wyłomańska A, Kowalczyk M. Groundwater Level Fluctuation Analysis in a Semi-Urban Area Using Statistical Methods and Data Mining Techniques—A Case Study in Wrocław, Poland. Applied Sciences. 2020; 10(10):3553. https://doi.org/10.3390/app10103553
  2. Mrozik, K., & Przybyła, Cz. (2013). An Attempt to Introduce Cultivation and Planning Measures into the Decision-Making Process in Order to Improve Water-Retaining Capacity of River Catchments. Polish Journal of Environmental Studies. 2013;22(6):1767-1773. http://www.pjoes.com/pdf-89145-23004?filename=An%20Attempt%20to%20Introduce.pdf
  3. Kochanek, K., & Tynan, S. (2010). The environmental risk assessment for decision support system for water management in the vicinity of open cast mines (DS WMVOC). Technological and Economic Development of Economy, 16(3), 414-431. https://doi.org/10.3846/tede.2010.26
  4. Hermanowski, P., & Ignaszak, T. (2016). Groundwater vulnerability based on four different assessment methods and their quantitative comparison in a typical North European Lowland river catchment (the Pliszka River catchment, western Poland). Geological Quarterly, 61(1), 166-176, doi: 10.7306/gq.1331. doi:https://doi.org/10.7306/gq.1331

Please describe also the main limitations of the proposed method?

Stay healthy and best regards

Author Response

Dear reviewer,

Thank you for your comments and suggestions. We appreciate your effort, so we considered all comments, and below we successively answered each one.

COMMENT 1: I do not understand why a municipality (community) for which there is no basic data was selected? There is also a lack of verification of the obtained results. If there is no available multi-year period data, at least single attempts in the field should be made.

  • ANSWER: We have chosen this area due to its spatial heterogeneity in a relatively small area which is representative of the whole Vojvodina province and other similar areas in the Pannonian basin. (p3. 104-110) Thus, generating good results we can apply this methodology in other similar geographic regions.
  • The lack of groundwater data but also other detailed geospatial data is a general problem for the whole Vojvodina region. There are no complete spatially and temporally continuous data series. This was the reason why we have used only six features and why we do not have verification. However, our aim was to provide new data, methods and directions for future research. We hope that it will help in setting up a dense network of measuring points for groundwater monitoring and enable more research about the groundwater regimes in this region.

COMMENT 2: I am still not satisfied with the discussion of results. But thanks to the changes I have noticed an increase in the scientific quality of the article. Perhaps it is worth expanding the introduction and discussion chapter a bit. Below are some examples from Central Europe that may be useful.

  1. Worsa-Kozak M, Zimroz R, Michalak A, Wolkersdorfer C, Wyłomańska A, Kowalczyk M. Groundwater Level Fluctuation Analysis in a Semi-Urban Area Using Statistical Methods and Data Mining Techniques—A Case Study in Wrocław, Poland. Applied Sciences. 2020; 10(10):3553. https://doi.org/10.3390/app10103553
  2. Mrozik, K., & Przybyła, Cz. (2013). An Attempt to Introduce Cultivation and Planning Measures into the Decision-Making Process in Order to Improve Water-Retaining Capacity of River Catchments. Polish Journal of Environmental Studies. 2013;22(6):1767-1773. http://www.pjoes.com/pdf-89145-23004?filename=An%20Attempt%20to%20Introduce.pdf
  3. Kochanek, K., & Tynan, S. (2010). The environmental risk assessment for a decision support system for water management in the vicinity of open cast mines (DS WMVOC). Technological and Economic Development of Economy16(3), 414-431. https://doi.org/10.3846/tede.2010.26
  4. Hermanowski, P., & Ignaszak, T. (2016). Groundwater vulnerability based on four different assessment methods and their quantitative comparison in a typical North European Lowland river catchment (the Pliszka River catchment, western Poland). Geological Quarterly, 61(1), 166-176, doi: 10.7306/gq.1331. doi:https://doi.org/10.7306/gq.1331

  • ANSWER: Thank you for your suggestions and research examples. We expanded introduction and discussion using this literature.

COMMENT 3: Please describe also the main limitations of the proposed method?

  • ANSWER: We described main limitations of the proposed method in Discussion section (line 358 – 546)

Best regards!

Reviewer 4 Report

The Authors used multi-criteria analysis to identify areas with good conditions for groundwater recharge. They modified the commonly used method of calculating the effective infiltration by changing the layer "depth to the groundwater table" into the layer "drainage density". This change was forced by the lack of hydrogeological research of the studied area, including the lack of data on the depth to the groundwater table.

1) The modification proposed by the Authors is interesting, but the lack of comparison with the results of calculations using the classical method does not allow to determine the influence of the modification on the calculation result.
2) I suggest excluding river and water bodies from the analysis area. Determining soil type and terain slope for rivers and water bodies is wrong.
3) I propose to change the class division of the "slope" layer. Research Sharma et all and Fox et all has shown that increasing the slope above 10 degrees does not alter the effective infiltration.
Fox D.M., Bryan R.B.,  Price A.G.    1997    The influence of slope angle on final infiltration rate for interrill conditions. Geoderma 80 (1-2): 181–194.    
Sharma K.D., Singh H.P., Pareek O.P.     1983    Rainwater infiltration into a bare loamy sand. Hydrological Sciences Journal 28 (3): 417-424.
4) Part of the studied area is in the valleys of large rivers. In such areas, evapotranspiration is greater than effective infiltration. This issue is not discussed in the article.

Author Response

Dear reviewer,

Thank you for your comments and suggestions. We appreciate your effort, so we considered all the comments and we successively answered on each one.

The Authors used multi-criteria analysis to identify areas with good conditions for groundwater recharge. They modified the commonly used method of calculating the effective infiltration by changing the layer "depth to the groundwater table" into the layer "drainage density". This change was forced by the lack of hydrogeological research in the studied area, including the lack of data on the depth of the groundwater table.

  • We used drainage density as an additional, very important feature for groundwater potential delineation.

COMMENT 1: The modification proposed by the Authors is interesting, but the lack of comparison with the results of calculations using the classical method does not allow to determine the influence of the modification on the calculation result.

  • ANSWER: In the previous versions of the manuscript, we used the classical AHP method and calculated GWPZ. All results were under review in the previous round. However, because of some limitations of the methodology and reviewers’ suggestions, we changed the methodology using extensive AHP - Fuzzy AHP. We reworked all processes and compared weights obtained by using AHP and FAHP. We added this in supplementary material which you can find attached with the paper. We showed the difference in applying both methodologies. After that, the final GWPZ map was calculated using FAHP weights.

COMMENT 2: I suggest excluding river and water bodies from the analysis area. Determining soil type and terrain slope for rivers and water bodies is wrong.

  • ANSWER: The suggestion to exclude water bodies and rivers from the research area was vey useful. We did it for soil type and slope because as you mentioned, the determination of soil type and slope is technically not possible for water bodies. For other factors, we left water bodies and rivers as relevant features because they are very important in terms of the continuous nature of recharging groundwater.

New maps and statistics for soil type, slope, and GWPZ were added in section 3.2.2 Soil, 3.2.5. Slope and 3.2.7. GWPZ.

COMMENT 3: I propose to change the class division of the "slope" layer. Research Sharma et all and Fox et all has shown that increasing the slope above 10 degrees does not alter the effective infiltration.
- Fox D.M., Bryan R.B.,  Price A.G.    1997    The influence of slope angle on final infiltration rate for interrill conditions. Geoderma 80 (1-2): 181–194.   

- Sharma K.D., Singh H.P., Pareek O.P.     1983    Rainwater infiltration into a bare loamy sand. Hydrological Sciences Journal 28 (3): 417-424.

  • ANSWER: We changed class divisions according to knowledge from Fox et al (1997) research. Previous 5 classes we reduced with 3 classes where the third class was dedicated to the slope larger than 11° (According to Fox, infiltration decreased until 11,5°) and one weight was assigned. New categorization, proposed methodology, new statistics, and new slope map are described in section 3.2.5. Slope.

COMMENT 4: Part of the studied area is in the valleys of large rivers. In such areas, evapotranspiration is greater than effective infiltration. This issue is not discussed in the article.

  • ANSWER: In this research, we did not consider data about evapotranspiration again because of the lack of relevant high spatial and temporal resolution meteorological time series data, that could be used in evapotranspiration calculation. According to the literature overview (some references are listed below), evapotranspiration is not considered in similar studies by other authors. However, in further research, it will be very useful to use the GWPZ map, ET, and other relevant data for the assessment of groundwater usage or some other important question.
  • Yıldırım, Ü. (2021). Identification of groundwater potential zones using GIS and multi-criteria decision-making techniques: a case study upper Coruh River basin (NE Turkey). ISPRS International Journal of Geo-Information, 10(6), 396.
  • Pathmanandakumar, V., Thasarathan, N., & Ranagalage, M. (2021). An Approach to Delineate Potential Groundwater Zones in Kilinochchi District, Sri Lanka, Using GIS Techniques. ISPRS International Journal of Geo-Information, 10(11), 730.
  • Chaudhry, A. K., Kumar, K., & Alam, M. A. (2021). Mapping of groundwater potential zones using the fuzzy analytic hierarchy process and geospatial technique. Geocarto International, 36(20), 2323-2344.
  • Gumma, M. K., & Pavelic, P. (2013). Mapping of groundwater potential zones across Ghana using remote sensing, geographic information systems, and spatial modeling. Environmental monitoring and assessment, 185(4), 3561-3579.
  • Kumar, M., Singh, P., & Singh, P. (2021). Fuzzy AHP based GIS and remote sensing techniques for the groundwater potential zonation for Bundelkhand Craton Region, India. Geocarto International, 1-24.
  • Shao, Z., Huq, M. E., Cai, B., Altan, O., & Li, Y. (2020). Integrated remote sensing and GIS approach using Fuzzy-AHP to delineate and identify groundwater potential zones in semi-arid Shanxi Province, China. Environmental Modelling & Software, 134, 104868.

Best regards!

Round 2

Reviewer 3 Report

Dear Authors,

Thanks for your effort and for introducing suggested changes.

In my opinion the manuscript Assessment of Groundwater Potential Zones Using GIS and AHP Techniques – A Case Study of the Titel Municipality (North Serbia) seems now suitable to be published in the International Journal of Geo-Information.

Best regards

Reviewer 4 Report

Dear Authors,

thank you for correcting the text in line with my comments.
In my opinion, the article can be published in present form. 

Kind regards

This manuscript is a resubmission of an earlier submission. The following is a list of the peer review reports and author responses from that submission.

Round 1

Reviewer 1 Report

Dear Authors,

Thanks for your effort. The topic under discussion is important, but in its current form, it will not arouse any broader interest. The work lacks a scientific level, especially the discussion of the results… Unfortunatelly in its current form, the article does not bring valuable conclusions to scientific knowledge. Here are some tips that could improve the quality of the article.

 The work structure is poor. The work lacks the basic elements of each scientific article, especially the discussion of the results. Also other parts are underdeveloped: literature review - introduction, methodology, description of results, conclusions ...

  1. The AHP approach combines elements of mathematics and psychology. Thanks to this, it is possible to solve decision problems that are multi-faceted and combine quantitative and qualitative elements. The approach proposed in the work uses typical and commonly used source data, related only to environmental aspects… Therefore, the use of AHP seems inadequate.
  2. The article should include a map with a hydrographic and drainage network.
  1. The study lacks information on local groundwater fluctuations
  2. What is the relationship of the analyzed commune (Titel) with the Vojvodina territory (province)? Please indicate on the map. Are there no more precise meteorological data from the commune area?
  3. In any scientific work, I would also expect a hypothesis. A well-formulated aim facilitates the description of the results and drawing conclusions.
  4. Figure 4. Geology abbreviations are not explained in the text. Perhaps it is worth introducing them additionally to Table 2.
  5. In the description of the results, the percentages should be consistently given (similar to description of land use or Table 3).
  6. The article lacks also statistical analysis of results…
  7. The conclusions are unclear and of little scientific importance. What are lessons learnt?

Best regards and stay healthy

Author Response

Dear reviewer,

Thank you for your comments and suggestions. We appreciate your effort, so we considered all the comments, and in the attached document we successively answered each one.

Sincerely

Reviewer 2 Report

This manuscript aims to conduct GIS and Analytic Hierarchy Process (AHP) techniques for delineation of the groundwater potential zones (GWPZ) in the Titel Municipality. Six thematic layers such as geology, geomorphology, LULC, drainage density, soil, and slope were used to delineate the GWPZ. Based on my review, the content is interested for publication, however some comments are suggested. The following comments are suggested to improve the quality of this work.

General comments:

  • Authors indicated that Groundwater is a very important renewable resource, but due to the drastic increase in population on the planet and increasing need for water resources, anthropogenic activities have a strong influence on discharging of groundwater reservoirs. But, how do these human activities affect discharging of groundwater reservoirs?
  • Authors used several parameters for the development of groundwater resources. However, the sensitivity of these parameters are necessary to be indicated.
  • In this study, authors integrated GIS and AHP techniques to find the potential groundwater zones in the small Municipality of Titel. However, how this integration process conducted? This point is necessary to clearly present in the revised version of the manuscript.
  • Based on the analysis of this study, six thematic layers such as geology, geomorphology, LULC, drainage density, soil, and slope were used to delineate the GWPZ. How was the data of these six layers obtained?
  • One key issue when you use AHP to do risk assessment is to collect the experts’ reply on questionnaire. In AHP, to determine fuzzy number, generally the is required. There are two approaches to do experts questionnaire: (1) one is the pairwise comparison, proposed by Saaty (1977), improved by Li et al (2013); (2) The second method to do questionnaire is to use table comparison proposed by Lyu et al (2020). Thus, please discuss how you did experts questionnaire? Which type of the experienced experts you invited? How do you determine the fuzzy number? Lyu et al's new questionnaire method can not only get the appropriate experts' reply but also can determine the fuzzy number based on experts' replies.
  • Saaty, T.L. (1977). "A scaling method for priorities in hierarchical structures." Journal of Mathematical Psychology, 15, 234-281.
  • Li, F., et al (2013). "Improved AHP method and its application in risk identification." Journal of Construction Engineering and Management, 10.1061/(ASCE)CO.1943-7862.0000605, 139(3), 312-320.
  • Lyu, H.M., et al (2020). Risk assessment using a new consulting process in fuzzy AHP. Journal of Construction Engineering and Management, ASCE, 146(3), 04019112. http://dx.doi.org/10.1061/(ASCE)CO.1943-7862.0001757
  • The technical writing of the manuscript is necessary to be double checked.

Specific comments

  • Authors need to present the main parameters that affects the delineation of groundwater potential zones based on literature.
  • Authors need to increase the visibility of introduction by presenting the recent studies, which
  • assessed the groundwater potential zones using different techniques, e.g., TOPSIS, TODIM etc.
  • The last section in introduction need to revise, please be systematic and present the main different between your research and current literature. Otherwise, what is the new in this work?
  • Line 80-100: the hydrological conditions of Vojvodina region should be clearly presented.
  • Line 141-142:
  • How were the Geomorphological map of the autonomous province of vojvodina used to obtain geomorphological data?
  • Why the authors used specifically six different thematic layers?
  • Authors need to clearly present the main geological parameters that effect on the occurrence and distribution of groundwater.
  • Geology map of the study area that presented in Fig. 4 is vague.
  • The definitions of all legends should be defined.
  • Line 290: the drainage density map of the Titel Municipality is presented in Fig. 7 but the discussion of this figure should be clearly provided.
  • Line 330: I recommend the authors to support the conclusions with more effective results.

Author Response

(The authors gave the same response as above.)

Reviewer 3 Report

The article is very interesting but needs to be improved. The comments are summarized below.

1. Please add a drawing of a hierarchical tree
Please provide results of comparisons according to Saaty's scale. For each level of the tree. Please prepare supplementary materials. 
2. Please describe how λmax value was calculated.
3. Please provide RCI- Random consistency index value, in the table. 
4. No references to the results of the AHP method in the summary
5. No discussion on the application of the AHP method
6. In the introduction too little information about the AHP method.

Author Response

(The authors gave the same response as above.)

Round 2

Reviewer 1 Report

Dear Authors,

Thank you for considering and introducing the suggested changes. I am still not satisfied with the discussion of results. But thanks to the changes I have noticed an increase in the scientific quality of the article. Perhaps it is worth expanding the introduction and discussion chapter a bit. Below are some examples from Poland that may be useful.

  1. Groundwater Level Fluctuation Analysis in a Semi-Urban Area Using Statistical Methods and Data Mining Techniques—A Case Study in Wrocław, Poland. Applied Sciences. 2020; 10(10):3553. https://doi.org/10.3390/app10103553
  2. An Attempt to Introduce Cultivation and Planning Measures into the Decision-Making Process in Order to Improve Water-Retaining Capacity of River Catchments. Polish Journal of Environmental Studies. 2013;22(6):1767-1773. http://www.pjoes.com/pdf-89145-23004?filename=An%20Attempt%20to%20Introduce.pdf
  3. The environmental risk assessment for decision support system for water management in the vicinity of open cast mines (DS WMVOC). Technological and Economic Development of Economy, 16(3), 414-431. https://doi.org/10.3846/tede.2010.26
  4. Groundwater vulnerability based on four different assessment methods and their quantitative comparison in a typical North European Lowland river catchment (the Pliszka River catchment, western Poland). Geological Quarterly, 61(1), 166-176, doi: 10.7306/gq.1331. doi:https://doi.org/10.7306/gq.1331

Stay healthy and best regards

Reviewer 2 Report

  • Please compare the result by use of the pairwise comparison (Saaty, 1977) and results by use of table comparison proposed by Lyu et al (2020).
  • How do you determine the fuzzy number? Lyu et al's new questionnaire method can not only get the appropriate experts' reply but also can determine the fuzzy number based on experts' replies.
  • Saaty, T.L. (1977). "A scaling method for priorities in hierarchical structures." Journal of Mathematical Psychology, 15, 234-281.
  • Lyu, H.M., et al (2020). Risk assessment using a new consulting process in fuzzy AHP. Journal of Construction Engineering and Management, ASCE, 146(3), 04019112. http://dx.doi.org/10.1061/(ASCE)CO.1943-7862.0001757
  • Research still has other flaws, innovation is weak, please add additional experiment data.

Reviewer 3 Report

Please add a drawing of a hierarchical tree.